# Baseline clinical characteristics and prognostic factors in hospitalized COVID-19 patients aged ≤ 65 years: A retrospective observational study

**Marta Betti**[1]*, **Marinella Bertolotti**[1], **Daniela Ferrante**[2], **Annalisa Roveta**[1], **Carolina Pelazza**[1], **Fabio Giacchero**[1], **Serena Penpa**[1], **Costanza Massarino**[1], **Tatiana Bolgeo**[1], **Antonella Cassinari**[1], **Marco Mussa**[3], **Guido Chichino**[3], **Antonio Maconi**[1]

1 Infrastruttura Ricerca Formazione Innovazione, Dipartimento Attività Integrate Ricerca Innovazione, Azienda Ospedaliera "SS Antonio e Biagio e Cesare Arrigo", Alessandria, Italy, 2 Unit of Medical Statistics, Department of Translational Medicine, University of "Piemonte Orientale" and Cancer Epidemiology, CPO Piemonte, Novara, Italy, 3 Infectious Diseases Unit, Azienda Ospedaliera "SS Antonio e Biagio e Cesare Arrigo", Alessandria, Italy

* marta.betti@ospedale.al.it

**Data Availability Statement:** All relevant data are within the manuscript and its Supporting Information files.

## Abstract

### Background

Individual differences in susceptibility to SARS-CoV-2 infection, symptomatology and clinical manifestation of COVID-19 have thus far been observed but little is known about the prognostic factors of young patients.

### Methods

A retrospective observational study was conducted on 171 patients aged ≤ 65 years hospitalized in Alessandria's Hospital from 1st March to 30th April 2020 with laboratory confirmed COVID-19. Epidemiological data, symptoms at onset, clinical manifestations, Charlson Comorbidity Index, laboratory parameters, radiological findings and complications were considered. Patients were divided into two groups on the basis of COVID-19 severity. Multivariable logistic regression analysis was used to establish factors associated with the development of a moderate or severe disease.

### Findings

A total of 171 patients (89 with mild/moderate disease, 82 with severe/critical disease), of which 61% males and a mean age (± SD) of 53.6 (± 9.7) were included. The multivariable logistic model identified age (50–65 vs 18–49; OR = 3.23 CI95% 1.42–7.37), platelet count (per 100 units of increase OR = 0.61 CI95% 0.42–0.89), c-reactive protein (CPR) (per unit of increase OR = 1.12 CI95% 1.06–1.20) as risk factors for severe or critical disease. The multivariable logistic model showed a good discriminating capacity with a C-index value of 0.76.

**Funding:** The authors received no specific funding for this work.

**Competing interests:** The authors have declared that no competing interests exist.

## Interpretation

Patients aged ≥ 50 years with low platelet count and high CRP are more likely to develop severe or critical illness. These findings might contribute to improved clinical management.

## Introduction

In December 2019, a novel coronavirus, severe acute respiratory syndrome coronavirus 2 (SARS-CoV-2; previously knowns as 2019-nCoV) caused an outbreak of a febrile respiratory illness in Wuhan city, Hubei Province, China [1]. On February 11[st], 2020 the World Health Organization (WHO) named this disease as new coronavirus disease 2019 (COVID-19) and on March 11[st], 2020 WHO declared it a pandemic [2]. Globally, there were more than 53 million confirmed cases with a mortality rate of 3% reported to the WHO [3].

The first case diagnosed with COVID-19 in Italy (at the Sacco Hospital in Milan) was confirmed by Istituto Superiore di Sanità (ISS) on February 21[st], 2020. Up through November 12[th], 2020 the total number of cases has risen by more than 1 million in Italy [4].

The clinical presentation of COVID-19 is that of a respiratory illness ranging from a mild common flu-like disease to a severe pneumonia. One of the largest cohort of patients with COVID-19 showed that most cases (81%) had mild to moderate illness, 14% had severe and 5% critical illness [5]. Patients with severe COVID-19 were likely to have comorbidities such as immunosuppression, diabetes and malignancy [6]. Elderly are more susceptible to SARS-CoV-2 and 70% of elderly COVID-19 patients are more likely to progress to severe or critical illness [7]. In patients aged ≥ 80 years low oxygen saturation at admission, high C-reactive protein and elevated lactate dehydrogenase are associated with rapid progression to death [8]. Recent data show that nearly 70% of people in the U.S. were younger than 60 years and the median age of U.S. COVID-19 patients was 48 years [9]. Although younger adults are less likely than older adults to develop severe infections, 5% of all severe cases aged ≤ 50 years develop serious symptoms and complications including severe pneumonia and, more rarely, encephalitis, cardiovascular disease [10].

So far, limited data are available for younger COVID-19 patients [11] thus, this study aims to describe the baseline clinical characteristics and highlight the prognostic factors of COVID-19 patients aged ≤ 65 years admitted to "SS Antonio e Biagio e Cesare Arrigo" Alessandria's Hospital in the Piedmont Region in northern Italy.

## Methods

### Study design and participants

This case series is part of COVID-19 Registry study, a single-center ambispective observational study carried out at "SS Antonio e Biagio e Cesare Arrigo" Alessandria's Hospital in the Piedmont Region in northern Italy. The study was approved by the Institutional Review Board (IRB) (protocol number ASO.IRFI.20.03) with a waiver informed consent for the retrospective part. The study was conducted in accordance with the Declaration of Helsinki and the Good Clinical Practices guidelines for observational studies. All consecutive adult patients aged between 18–65 years admitted to the Alessandria's Hospital with laboratory-confirmed COVID-19 from March 1[st] to April 30[th] were enrolled. Patients discharged from Emergency Department were excluded.

## Procedures

In order to detect virus infection, SARS-CoV-2 nucleic acid was identified by nasopharyngeal swab specimen in all patients by real-time reverse-transcriptase polymerase chain reaction (RT-PCR), according to WHO laboratory guidance [12].

Data of hospitalized patients was extracted from electronic medical records system (TrackCare) and paper based medical records and entered by clinical study coordinators in a dedicated electronic case report form (eCRF), specifically developed using the freely available Research Electronic Data Capture (*REDCap*) platform [13]. All data was pseudonymed according to clinical study and data protection regulations. The collected variables were the following: demographical characteristics (age, sex, residence), symptoms at onset (fever, chills, cough, conjunctivitis, rhinorrhea, headache, muscle pain, fatigue, nausea, vomiting, diarrhea, dyspnea, hemoptysis, hematemesis, ageusia, anosmia), clinical manifestations (temperature, blood pressure, oxygen saturation, heart rate, respiratory rate), comorbidities (acute myocardial infarction, chronic heart failure, peripheral vascular disease, dementia, Chronic Obstructive Pulmonary Disease (COPD), cerebrovascular disease, connective tissue disease, peptic ulcer disease, mild liver disease, diabetes, hemiplegia, moderate-to-severe kidney disease, diabetes with chronic complications, cancer without metastasis, leukemia, lymphoma, moderate or severe liver disease, metastasis, Acquired Immune-Deficiency Syndrome (AIDS)), laboratory parameters (hematology, biochemistry, blood coagulation, inflammatory markers), radiologic findings based on chest X-ray or CT scan (normal, monolateral or bilateral ground-glass opacity, interstitial involvement, irregular shading), complications (septic shock, acute respiratory distress syndrome, acute kidney injury, hemorrhages, rhabdomyolysis, pneumonia, heart failure, respiratory decompensation, hypoproteinemia, acidosis, sepsis, acute heart damage, pulmonary embolism, deep vein thrombosis, respiratory failure), treatment typologies (antibiotics, antivirals, antifungals, corticosteroids, oxygen therapy, non-invasive mechanical ventilation, invasive mechanical ventilation, ExtraCorporeal Membrane Oxygenation (ECMO), immunoglobulins, renal replacement therapy, chloroquine/hydroxychloroquine, antithrombotic prophylaxis, deep vein thrombosis/pulmonary embolism therapy), outcome (hospital discharge, transfer, death). For each patient, we calculated the Charlson Comorbidity Index (CCI) based on the available data [14]. Patient's medical records were accessed until July 2020.

## Outcomes

According to WHO interim guidance [15], patients were divided into two groups: patients with mild or moderate disease and patients with severe or critical disease (Table 1).

**Table 1. Classification of COVID-19 disease severity.**

| Group 1 | | Group 2 | |
|---|---|---|---|
| *Mild disease* | *Moderate disease* | *Severe disease* | *Critical disease* |
| • Symptoms as fever, cough, fatigue, dyspnea, myalgia, headache, diarrhea, nausea/vomiting, loss of smell/taste <br> • No pneumonia | • X-ray findings of pneumonia <br> Blood oxygen saturation levels (SpO$_2$) ≥ 90% on room air <br> • No complications related to severe conditions | Mild or moderate clinical features plus at least one of the following manifestations: | Severe manifestations plus any other features that suggest disease rapid progression: |
| | | • respiratory rate > 30; <br> • breaths/minute or severe respiratory distress; <br> SpO$_2$ < 90% on room air | • respiratory failure with need mechanical ventilation; <br> • presence of acute respiratory distress syndrome (ARDS), sepsis, or septic shock, other complications include acute pulmonary embolism, acute coronary syndrome, acute stroke, and delirium |

## Statistical analysis

A forward multivariable logistic regression analysis was applied to the data in order to develop a predictive model for severe disease. Variables identified from the univariable analysis as potential predictors have been included in the multivariable analysis. A two-sided *P* value < 0.05 was considered statistically significant.

To validate the variables selection for the prognostic model, a bootstrap analysis was computed. Bootstrap-corrected Harrell's C-index was used to assess the model's discriminatory ability. The C-index was estimated by bootstrapping with 1000 resamples to estimate an unbiased measure of the ability of our predictive model to discriminate among patients with respect to their severe/not severe disease. Calibration was tested using the Hosmer-Lemeshow test. All analyses were performed using the programme *SAS* (Statistical Analysis Software *9.4*, *SAS* Institute Inc, Cary, North Carolina, USA) and STATA v.16 (*Stata* Corp, Texas, USA).

## Results

Between March 1$^{st}$ to April 30$^{th}$, 2020, 501 COVID-19 patients with laboratory-confirmed SARS-CoV-2 infection were hospitalized in the "SS Antonio e Cesare Arrigo" Hospital in Alessandria, Piedmont region (Italy), of whom 171 (34%) aged $\leq$ 65 years. The majority of young patients were males (61%) with a mean ($\pm$ SD) age of 53.6 ($\pm$ 9.7). On the basis of clinical manifestations 171 patients were divided into two groups: 89 cases with mild or moderate disease (group 1) and 82 cases with severe or critical disease (group 2) (Tables 2 and 3). Patients were mainly male both in group 1 (56.2%) and in group 2 (65.9%). The mean ($\pm$ SD) age was higher in the group 2 (56.72 $\pm$ 6.75) than group 1 (50.85 $\pm$ 11.2). In both groups, the majority of patients aged $\geq$ 50 years (group 1, 60.7%; group 2, 85.4%) and presented more than one symptoms at onset (group 1, 73%; group 2, 78.1%). Charlson Comorbidity Index (CCI) showed a higher score in patients with severe or critical disease than in patients with mild or moderate disease (group 2, 1.98; group 1, 1.79). Patients aged $\geq$ 50 years are more likely to develop severe or critical illness (85.4% versus 60.7%, $p$<0.0001). There were no significant differences in the distribution of sex, symptoms at onset, and Charlson Comorbidity Index between the two groups.

As shown in Table 3, comparison of laboratory parameters between the two groups demonstrates that patients with severe or critical disease have a significant decrease of lymphocytes ($p$ = 0.001) and calcium ($p$ = 0.006). The same patients have a significant increase of D-dimer ($p$ = 0.021), LDH ($p$<0.0001), C-reactive protein ($p$<0.0001), ferritin ($p$<0.0001), urea ($p$ = 0.013) and fibrinogen ($p$ = 0.044).

Finally, multivariable logistic model confirmed age $\geq$ 50 years (50–65 vs 18–49; $p$ = 0.005, OR = 3.23 CI95% 1.42–7.37) and high c-reactive protein (CPR) level (per unit of increase, $p$<0.0001, OR = 1.12 CI95% 1.06–1.20) as risk factors for severe or critical disease. Additionally, we found that patients in the severe or critical group showed a platelet count significantly lower (per 100 units of increase $p$ = 0.010, OR = 0.61 CI95% 0.42–0.89) than those in the mild or moderate group (Table 4).

The multivariable logistic model showed a good discriminating capacity with a C-index value of 0.76 (Fig 1) and a bias-corrected C-index equal to 0.75. The calculated chi-square statistics for calibration is $\chi 2$ = 8.60; $p$ = 0.38 which indicated goodness-of-fit of the model.

## Discussion

The present case series evaluated the clinical characteristics and prognostic factors of hospitalized patients aged $\leq$ 65 years affected by COVID-19. The difference between patients with mild or moderate disease and patients with severe or critical disease were analyzed. Patients

**Table 2. Epidemiological and baseline clinical features of hospitalized COVID-19 patients aged ≤ 65 years divided into two groups according to clinical manifestations (Group 1 –mild and moderate disease; Group 2 –severe and critical disease): Univariable logistic regression analysis.**

| Variable | Mild/moderate disease[§] (n = 89) | Severe/critical disease° (n = 82) | p-value |
|---|---|---|---|
| **Epidemiological data** | | | |
| Female, n° (%) | 39 (43.8%) | 28 (34.1%) | |
| Male, n° (%) | 50 (56.2%) | 54 (65.9%) | 0.196 |
| Age (years), mean ± SD | 50.85 ± 11.2 | 56.72 ± 6.75 | |
| Age <50 years | 35 (39.3%) | 12 (14.6%) | |
| Age ≥50 years | 54 (60.7%) | 70 (85.4%) | **<0.0001** |
| **Symptoms at onset, n° (%)** | | | |
| n° ≤1 | 24 (27%) | 18 (21.9%) | |
| n° >1 | 65 (73%) | 64 (78.1%) | 0.447 |
| **Comorbidities** | | | |
| Charlson Comorbidity Index^, mean ± SD | 1.79 ± 2.3 | 1.98 ± 1.61 | 0.536 |
| Acute myocardial infarction, n° (%) | 4 (4.5%) | 3 (3.6%) | |
| Chronic heart failure, n° (%) | 4 (4.5%) | 0 | |
| Peripheral vascular disease, n° (%) | 4 (4.5%) | 1 (1.2%) | |
| Dementia, n° (%) | 0 | 4 (4.9%) | |
| Chronic obstructive pulmonary disease (COPD), n° (%) | 3 (3.4%) | 5 (6.1%) | |
| Cerebrovascular disease, n° (%) | 2 (2.2%) | 3 (3.6%) | |
| Connective tissue disease, n° (%) | 1 (1.1%) | 2 (2.4%) | |
| Peptic ulcer disease, n° (%) | 1 (1.1%) | 1 (1.2%) | |
| Mild liver disease, n° (%) | 3 (3.4%) | 0 | |
| Diabetes, n° (%) | 6 (6.7%) | 4 (4.9%) | |
| Hemiplegia, n° (%) | 4 (4.5%) | 0 | |
| Moderate-to-severe kidney disease, n° (%) | 4 (4.5%) | 5 (6.1%) | |
| Diabetes with chronic complications, n° (%) | 4 (4.5%) | 2 (2.4%) | |
| Cancer without metastasis, n° (%) | 3 (3.4%) | 6 (7.3%) | |
| Leukemia, n° (%) | 1 (1.1%) | 1 (1.2%) | |
| Lymphoma, n° (%) | 2 (2.2%) | 0 | |
| Metastasis, n° (%) | 5 (5.6%) | 1 (1.2%) | |

[§],°at admission to the Hospital. For Charlson Comorbidity Index data were available for n = 78 patients in both groups.

^The score on the Charlson Comorbidity Index is calculated on the basis of the presence of 19 conditions, each of which is assigned a weighted score of 1, 2, 3, or 6.

Higher scores indicate more coexisting conditions and a higher risk of death.

A two-sided *p value* < 0.05 was considered statistically significant.

aged ≤ 65 years with severe or critical illness represented 16% of all hospitalized COVID-19 patients. The findings of our study show that patients with severe or critical illness were more males than females and were older than patients with mild or moderate disease. Our results suggest that subjects in the age range which goes from 50 to 65 years old are more likely to develop a severe or critical COVID-19. Laboratory analysis at admission indicated that D-Dimer and C-reactive protein (CRP) levels were significantly higher in 82 patients with severe or critical disease than in 89 patients with mild or moderate disease. It is reasonable that these factors could predict COVID-19 severity in young patients. So far, it has been shown that CRP levels were correlated with severe SARS-CoV-2 infection in elderly patients [8, 11]. Moreover, severe COVID-19 has been shown to be associated with high D-dimer levels, which appear to predict mortality [16, 17]. High D-dimer and fibrinogen levels were also indicative of an increase in coagulation and consequently in venous thromboembolism [18–20]. In our

**Table 3. Laboratory parameters at admission of hospitalized COVID-19 patients aged ≤ 65 years divided into two groups according to clinical manifestations (Group 1 –mild and moderate disease; Group 2 –severe and critical disease): Univariable logistic regression analysis.**

| Laboratory findings Median (IQR) | Mild/moderate disease[§] (n = 89) | Severe/critical disease[°] (n = 82) | |
|---|---|---|---|
| Leukocytes count (10³/mcL) | 6.09 (4.46–8.14) | 6.77 (4.82–9.57) | 0.436 |
| Neutrophils count (10³/mcL) | 4.42 (2.89–5.94) | 5.55 (3.65–7.59) | 0.150 |
| Lymphocytes count (10³/mcL) | 1.03 (0.76–1.46) | 0.86 (0.66–1.12) | **0.001** |
| Erythrocytes count (10³/mcL) | 4.69 (4.25–5.05) | 4.87 (4.41–5.24) | 0.333 |
| Hemoglobin (g/dl) | 13.5 (12.15–14.8) | 14.1 (12.4–15.3) | 0.349 |
| Platelets (10³/mcL) | 206 (156–269) | 196 (149–243) | 0.255 |
| Hematocrit (%) | 40.6 (37.2–44.5) | 42.65 (38–45.3) | 0.470 |
| Eosinophils count (10³/mcL) | 0.02 (0.01–0.05) | 0.01 (0–0.03) | 0.417 |
| Prothrombin time (PT) (seconds) | 13.9 (13.05–15.1) | 13.9 (13.2–14.8) | 0.505 |
| Activated partial thromboplastin time (PTT) (seconds) | 30.7 (29.1–32.7) | 30.6 (28.7–33.1) | 0.961 |
| Ratio | 1.04 (0.99–1.11) | 1.04 (0.97–1.14) | 0.976 |
| D-dimer (mcg/mL) | 0.55 (0.45–0.88) | 1.23 (0.76–1.9) | **0.021** |
| Prothrombin time (International Normalised Ratio, INR) (ratio) | 1.03 (0.96–1.12) | 1.03 (0.98–1.1) | 0.487 |
| Sodium level (mEq/L) | 138 (136–140) | 138 (136–139) | 0.930 |
| Potassium level (mEq/L) | 4.2 (3.9–4.6) | 4.15 (3.8–4.4) | 0.240 |
| Chlorine level (mEq/L) | 102 (98–104) | 102 (97–104) | 0.424 |
| Calcium level (mg/dL) | 8.8 (8.5–9.1) | 8.4 (8–8.8) | **0.006** |
| LDH (U/L) | 503 (417–683) | 734.5 (597–998) | **<0.0001** |
| AST (U/L) | 25.5 (18–41.5) | 42 (30–56) | 0.350 |
| ALT (U/L) | 25 (17–41) | 31 (22–50) | 0.068 |
| Total bilirubin (mg/dL) | 0.56 (0.42–0.75) | 0.52 (0.39–0.74) | 0.420 |
| C-reactive protein (mg/dL) | 3.58 (1.31–8.28) | 10.13 (4.07–16.53) | **<0.0001** |
| Ferritin (ng/mL) | 352.3 (154.6–783.7) | 1020 (557.3–1486) | **<0.0001** |
| Creatinine (mg/dL) | 0.76 (0.6–0.87) | 0.88 (0.72–1.03) | 0.105 |
| Troponin (ng/L) | 6 (2–11) | 10 (4–29) | 0.337 |
| Urea (mg/dL) | 30 (24–39) | 39 (30–60) | **0.013** |
| Direct Bilirubin (mg/dL) | 0.2 (0.14–0.29) | 0.2 (0.17–0.35) | 0.563 |
| Gamma glutamyl transferase (GGT) (U/L) | 35.5 (21–80) | 58.5 (34.5–110) | 0.139 |
| Fibrinogen (mg/dL) | 536 (413–673) | 625 (557.5–726.5) | **0.044** |
| Prothrombin (%) | 93 (77.5.-102.5) | 92 (85–103) | 0.568 |

[§]For each variable data were available for (n =): Leukocytes (n = 88), Neutrophils (n = 88), Lymphocytes (n = 88), Erythrocytes (n = 88), Hemoglobin (n = 88), Platelets (n = 87), Hematocrit (n = 87), Eosinophils (n = 88), Prothrombin time (n = 68), Activated partial thromboplastin time (n = 55), Ratio (n = 52), D-dimer (n = 49), INR (n = 68), Sodium (n = 86), Potassium (n = 83), Chlorine (n = 43), Calcium (n = 35), LDH (n = 73), AST (n = 52), ALT (n = 81), Total bilirubin (n = 61), C-reactive protein (n = 86), Ferritin (n = 71), Creatinine (n = 83), Troponin (n = 41), Urea (n = 35), Direct Bilirubin (n = 56), Gamma glutamyl transferase (n = 36), Fibrinogen (n = 25), Prothrombin (n = 36).

[°]For each variable data were available for (n =): Leukocytes (n = 82), Neutrophils (n = 82), Lymphocytes (n = 82), Erythrocytes (n = 82), Hemoglobin (n = 82), Platelets (n = 82), Hematocrit (n = 82), Eosinophils (n = 82), Prothrombin time (n = 72), Activated partial thromboplastin time (n = 57), Ratio (n = 51), D-dimer (n = 46), INR (n = 72), Sodium (n = 81), Potassium (n = 78), Chlorine (n = 35), Calcium (n = 34), LDH (n = 64), AST (n = 39), ALT (n = 75), Total bilirubin (n = 42), C-reactive protein (n = 76), Ferritin (n = 65), Creatinine (n = 76), Troponin (n = 41), Urea (n = 47), Direct Bilirubin (n = 41), Gamma glutamyl transferase (n = 24), Fibrinogen (n = 32), Prothrombin (n = 34).

case series fibrinogen was significantly higher in patients with severe or critical disease than patients with mild or moderate disease whereas no differences were found in prothrombin time but few data were available. Of 82 critically ill patients, three had a history of both pulmonary embolism and deep vein thrombosis and five showed only pulmonary embolism during hospitalization.

**Table 4. The relationship between characteristics on admission of patients with SARS-CoV-2 infection and severe/critical disease: Multivariable logistic regression analysis.**

| Variable | OR (95%CI) |
|---|---|
| Age, < 50 vs ≥ 50 years | 3.23 (1.42–7.37) |
| Platelets (per 100 units of increase) | 0.61 (0.42–0.89) |
| C-reactive protein (per unit of increase) | 1.12 (1.06–1.20) |

In our findings, a significant reduction of lymphocytes count was observed in patients with severe or critical disease at admission. A lymphocytopenia (count less than ≤ 1.0 x $10^9$/L as reported by Guan and colleagues) [21] was reported in 53% of all patients (45% group 1, 61% group 2). Lymphocytopenia is one of the most important hematological features in COVID-19 [21]. It is known that lymphocytopenia can be associated with progression disease [22] and increased mortality [7].

In laboratory findings, we also observed that an increment of per 100x$10^3$/mcL in platelet count was significantly associated with a 40% decrease in severity disease. Several studies [7, 23, 24] have reported that low platelet count is associated with increased risk of severe disease and mortality in patients with COVID-19, and thus should serve to discriminate between severe and non-severe COVID-19 infections.

Other biomarkers of COVID-19 severity include lactate dehydrogenase (LDH) and serum levels of ferritin [25]. Elevated LDH levels in COVID-19 patients were associated with death [8, 18] and critical illness in elderly patients [7]. The present data show that LDH was significantly higher in patients with severe or critical disease considering the univariable logistic regression analysis. The meta-analysis of 10 studies demonstrated that ferritin level was associated with mortality and severe COVID-19 patients [26]. Our data showed a significantly increased of serum levels of ferritin in patients with severe or critical disease.

Albumin, urea and creatinine are clinical indicators of both kidney function and disease progression [27]. In our laboratory findings, only urea showed a significantly increased in patients with severe COVID-19.

Comparing electrolytes (sodium, potassium, chloride and calcium) between COVID-19 patients with and without severe disease only calcium was significantly lower in patients with severe COVID-19. A pooled analysis of five studies established COVID-19 severity is associated with lower serum concentrations of sodium, potassium and calcium [28].

Our study has several limitations. The sample size was small but most results were consistent with other studies on elderly patients with COVID-19. The study was carried out at a single center, thus the results may not be an exact representation of the general population, so we are planning to include the data of other centers for additional analyses. Since our case series derived from a retrospective study some relevant clinical data, such as IL6, fibrinogen, prothrombin, procalcitonin and creatin kinase were not available or available for few patients. Although several studies identified strong correlation between severe COVID-19 and cardiovascular and metabolic diseases [29, 30], our study didn't find any statistically significant differences between these comorbidities and mild/moderate and severe/critical disease groups. Furthermore, this study didn't evaluate the mortality rate in patients with severe COVID-19, thus a larger case series should be considered for further analysis.

To the best of our knowledge, this is the first study in which the clinical features were compared between two main clinical presentations in a young adult population (defined as 18 to 65 years old) of COVID-19 patients.

In conclusion, our findings highlight clinical features of patients infected with SARS-CoV2 and show that age ≥ 50 years, C-reactive protein levels and platelets count were independent

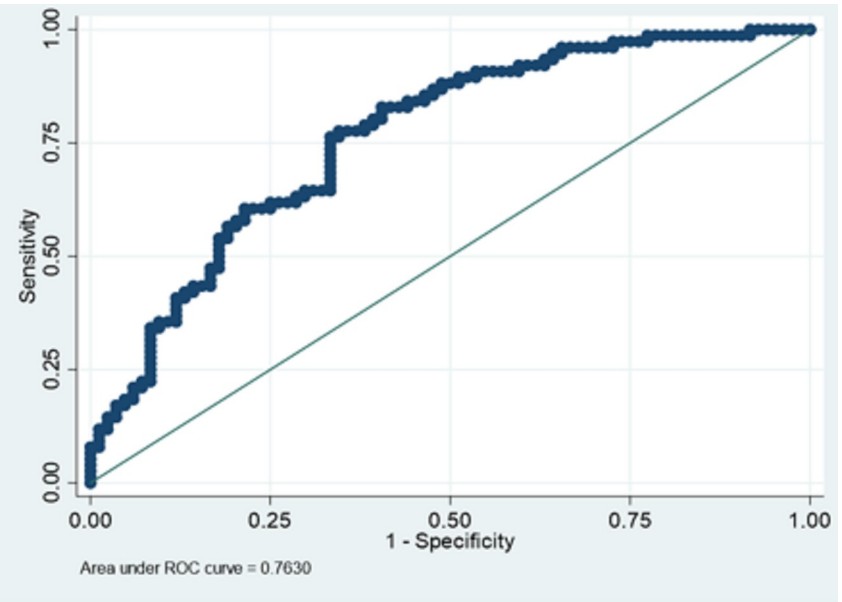

**Fig 1. ROC curve of multivariable logistic regression model.**

prognostic factors of disease severity in COVID-19 patients aged ≤ 65 years. These predictor factors could help the clinicians to identify high-risk hospitalized patients in order to improve clinical management of the pandemic also among younger subjects.

## Supporting information

**S1 Dataset.**
(XLSX)

## Author Contributions

**Conceptualization:** Fabio Giacchero, Guido Chichino, Antonio Maconi.

**Data curation:** Marta Betti, Daniela Ferrante, Annalisa Roveta, Carolina Pelazza, Fabio Giacchero, Serena Penpa, Costanza Massarino, Tatiana Bolgeo, Antonella Cassinari.

**Formal analysis:** Daniela Ferrante.

**Methodology:** Marta Betti, Marinella Bertolotti, Daniela Ferrante, Annalisa Roveta, Carolina Pelazza, Fabio Giacchero, Serena Penpa, Tatiana Bolgeo.

**Project administration:** Marta Betti, Marinella Bertolotti.

**Supervision:** Marinella Bertolotti, Annalisa Roveta, Guido Chichino, Antonio Maconi.

**Writing – original draft:** Marta Betti, Daniela Ferrante, Marco Mussa.

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
