## [Decision Letter · Decision Letter 0]

3 Feb 2021

PONE-D-20-41084

Baseline clinical characteristics and prognostic factors in hospitalized COVID-19 patients aged ≤ 65 years: a retrospective observational study

PLOS ONE

Dear Dr. Betti,

Thank you for submitting your manuscript to PLOS ONE. After careful consideration, we feel that it has merit but does not fully meet PLOS ONE’s publication criteria as it currently stands. Therefore, we invite you to submit a revised version of the manuscript that addresses the points raised during the review process.

The manuscript is interesting.  Provided you address the changes recommended, the manuscript will be accepted for publication. 

We look forward to receiving your revised manuscript.

Kind regards,

Prof. Raffaele Serra, M.D., Ph.D

Academic Editor

PLOS ONE

Journal Requirements:

2. In the ethics statement in the manuscript and in the online submission form, please provide additional information about the patient records/samples used in your retrospective study, including: a) whether all data were fully anonymized before you accessed them; b) the date range (month and year) during which patients' medical records/samples were accessed.

4.In your Data Availability statement, you have not specified where the minimal data set underlying the results described in your manuscript can be found. PLOS defines a study's minimal data set as the underlying data used to reach the conclusions drawn in the manuscript and any additional data required to replicate the reported study findings in their entirety. All PLOS journals require that the minimal data set be made fully available. For more information about our data policy, please see http://journals.plos.org/plosone/s/data-availability.

Additional Editor Comments:

After careful evaluation this manuscript should be revised according to the reviewers suggestions and resubmitted.

Reviewers' comments:

Reviewer's Responses to Questions

**Comments to the Author**

1. Is the manuscript technically sound, and do the data support the conclusions?

Reviewer #1: Yes

2. Has the statistical analysis been performed appropriately and rigorously? 

Reviewer #1: Yes

3. Have the authors made all data underlying the findings in their manuscript fully available?

Reviewer #1: Yes

4. Is the manuscript presented in an intelligible fashion and written in standard English?

Reviewer #1: Yes

5. Review Comments to the Author

Reviewer #1: The authors aimed to characterize Individual differences in susceptibility to SARS-CoV-2 infection, symptomatology and related clinical manifestations.

The paper is overall well written but in the Discussion section I would deepen a little bit the issues on cardiovascular disease and metabolic disease and covid-19 citing and discussing recent articles on these topics such as:

Ielapi N, et al. Cardiovascular disease as a biomarker for an increased risk of COVID-19 infection and related poor prognosis. Biomark Med. 2020;14(9):713-716.

Javanmardi F, et al. Prevalence of underlying diseases in died cases of COVID-19: A systematic review and meta-analysis. PLoS One. 2020 Oct 23;15(10):e0241265.

Moazzami B, et al. Metabolic risk factors and risk of Covid-19: A systematic review and meta-analysis. PLoS One. 2020 Dec 15;15(12):e0243600. doi: 10.1371/journal.pone.0243600. PMID: 33320875.

6. PLOS authors have the option to publish the peer review history of their article (what does this mean?). If published, this will include your full peer review and any attached files.

Reviewer #1: No

---

## [Author Response · Author response to Decision Letter 0]

3 Mar 2021

Reply to Reviewer #1:

Comment

The authors aimed to characterize Individual differences in susceptibility to SARS-CoV-2 infection, symptomatology and related clinical manifestations.

The paper is overall well written but in the Discussion section I would deepen a little bit the issues on cardiovascular disease and metabolic disease and covid-19 citing and discussing recent articles on these topics such as:

Ielapi N, et al. Cardiovascular disease as a biomarker for an increased risk of COVID-19 infection and related poor prognosis. Biomark Med. 2020;14(9):713-716.

Javanmardi F, et al. Prevalence of underlying diseases in died cases of COVID-19: A systematic review and meta-analysis. PLoS One. 2020 Oct 23;15(10):e0241265.

Moazzami B, et al. Metabolic risk factors and risk of Covid-19: A systematic review and meta-analysis. PLoS One. 2020 Dec 15;15(12):e0243600. doi: 10.1371/journal.pone.0243600. PMID: 33320875.

Reply

According to the reviewer's suggestion, we implemented the discussion on cardiovascular and metabolic diseases and COVID-19 by citing and discussing the suggested articles.

We have cited the article by Ielapi et al at page 14, line 14.

In the Discussion paragraph (page 15, lines 243-247) we have added the following sentence: “Although several studies identified strong correlation between severe COVID-19 and cardiovascular and metabolic diseases, our study didn’t find any statistically significant differences between these comorbidities and mild/moderate and severe/critical disease groups. Furthermore, this study didn’t evaluate the mortality rate in patients with severe COVID-19, thus a larger case series should be considered for further analysis.”

---

## [Editor Report · Decision Letter 1]

8 Mar 2021

Baseline clinical characteristics and prognostic factors in hospitalized COVID-19 patients aged ≤ 65 years: a retrospective observational study

PONE-D-20-41084R1

Dear Dr. Betti,

We’re pleased to inform you that your manuscript has been judged scientifically suitable for publication and will be formally accepted for publication once it meets all outstanding technical requirements.

Kind regards,

Prof. Raffaele Serra, M.D., Ph.D

Academic Editor

PLOS ONE

Additional Editor Comments (optional):

amended manuscript is acceptable.
---

## [Editor Report · Acceptance letter]

16 Mar 2021

PONE-D-20-41084R1 

Baseline clinical characteristics and prognostic factors in hospitalized COVID-19 patients aged ≤ 65 years: a retrospective observational study 

Dear Dr. Betti:

I'm pleased to inform you that your manuscript has been deemed suitable for publication in PLOS ONE. Congratulations! Your manuscript is now with our production department. 

Kind regards, 

on behalf of

Prof. Raffaele Serra 

Academic Editor

PLOS ONE